# A Cross-Sectional Validation of Horos and CoreSlicer Software Programs for Body Composition Analysis in Abdominal Computed Tomography Scans in Colorectal Cancer Patients

**DOI:** 10.3390/diagnostics14151696

**Published:** 2024-08-05

**Authors:** Andrés Jiménez-Sánchez, María Elisa Soriano-Redondo, José Luis Pereira-Cunill, Antonio Jesús Martínez-Ortega, José Ramón Rodríguez-Mowbray, Irene María Ramallo-Solís, Pedro Pablo García-Luna

**Affiliations:** 1Unidad de Gestión Clínica de Endocrinología y Nutrición, Instituto de Biomedicina de Sevilla, IBiS/Hospital Universitario Virgen del Rocío/CSIC/Universidad de Sevilla, Avda. Manuel Siurot s/n, 41013 Seville, Spain; 2Unidad de Gestión Clínica de Radiodiagnóstico, Hospital Universitario Virgen del Rocío, Avda. Manuel Siurot s/n, 41013 Seville, Spain; 3Unidad de Gestión Clínica de Oncología Médica, Hospital Universitario Virgen del Rocío, Avda. Manuel Siurot s/n, 41013 Seville, Spain; 4Unidad de Gestión Clínica de Cirugía General y del Aparato Digestivo, Hospital Universitario Virgen del Rocío, Avda. Manuel Siurot s/n, 41013 Seville, Spain

**Keywords:** Alberta protocol, Horos, CoreSlicer, computed tomography, body composition, muscle mass, colorectal cancer, sarcopenia

## Abstract

Background: Body composition assessment using computed tomography (CT) scans may be hampered by software costs. To facilitate its implementation in resource-limited settings, two open-source segmentation programs (Horos and CoreSlicer) were transversally validated in colorectal cancer patients. Methods: Contrast-enhanced abdominal CT scans were analyzed following the Alberta protocol. The Cross-Sectional Area (CSA) and intensities of skeletal muscle tissue (MT), subcutaneous adipose tissue (SAT), visceral adipose tissue (VAT), and intramuscular adipose tissue (IMAT) were measured. The Skeletal Muscle Index (SMI) was calculated. Cutoff points were applied to the SMI, MT intensity, and VAT CSA to define muscle atrophy, myosteatosis, and abdominal obesity. The inter-software agreement was evaluated using different statistical tools. Results: A total of 68 participants were measured. The MT CSA and SMI displayed no differences. The MT CSA agreement was excellent, and both programs provided equal muscle atrophy prevalences. CoreSlicer underestimated the MT intensity, with a non-significant myosteatosis prevalence increase (+5.88% and +8.82%) using two different operative definitions. CoreSlicer overestimated the CSA and intensity in both VAT and SAT, with a non-significant increase (+2.94%) in the abdominal obesity prevalence. Conclusions: Both software programs were feasible tools in the study group. The MT CSA showed great inter-software agreement and no muscle atrophy misdiagnosis. Segmentation differences in the MT intensity and VAT CSA caused limited diagnostic misclassification in the study sample.

## 1. Introduction

Computed tomography (CT) is considered the reference technique for body composition analysis in oncology [1], as it is an indirect technique with high spatial resolution, accuracy, and reproducibility [2]. Like magnetic resonance imaging (MRI), these imaging techniques can determine fatty infiltration in the muscle (myosteatosis) and measure visceral adipose tissue (VAT). Logistically, CT scans allow for opportunistic or retrospective measurements in routine studies requested in medical or surgical services for diagnostic–therapeutic purposes. As drawbacks, it is a high-cost and ionizing technique, although this could change with the introduction of low-radiation protocols for body composition analysis. Measurements are regional, yet whole-body estimation models are available [3,4]. Regarding technical error, the presence of intravenous contrast [5], volume overload [6], slice thickness, and tube current [7,8] should be taken into account.

Image segmentation allows for the quantitative measurement of the Cross-Sectional Area (CSA, usually expressed in cm^2^) of tissues in a region of interest (ROI). This process is based on the unique radiation absorption of each tissue, expressed as the attenuation intensity in Hounsfield Units (HU), and an adequate location of anatomical landmarks. An archetypical image for this task is an axial slice located in the third lumbar vertebra (L3), since it is the abdominal location with the maximum individual representativeness and interindividual variability [9]. The following tissues can be segmented for body composition analysis at this location: muscle tissue (MT), subcutaneous adipose tissue (SAT), VAT, and intramuscular adipose tissue (IMAT).

Manual or semi-automatic image segmentation can be a labor-intensive and operator-dependent procedure. To overcome this barrier, artificial intelligence (AI)-based software programs for body composition analysis allow for fully automated tissue segmentation [10], dramatically speeding up this process [11,12]. This opens the possibility of performing a 3D analysis of the body composition, analyzing all the images in the study in a fast and feasible way. These AI-based programs usually have restricted access and also have technical limitations: although their performance can be excellent, cases of erroneous segmentation can still occur [11]. For the time being, 2D and human-guided analyses still have a place.

In colorectal cancer patients, the estimation of muscle atrophy using CT scans has been shown to be an independent predictor of events of interest such as survival [1,13,14,15,16,17]; physical, cognitive and social functionality [18,19]; quality of life [20]; postoperative complications [21]; length of hospital stay [22,23]; and the need for in-hospital rehabilitation or discharge to a nursing home [24]. Due to methodological heterogeneity, previous studies in colorectal cancer calculated a prevalence of muscle atrophy ranging from 15 to 60% and of myosteatosis from 19 to 78% [25]. Myosteatosis in CT scans has also been associated with a reduced survival time in digestive malignancies [15,16,17,26,27,28], increased risk of post-surgical complications [29], and reduced physical function [19]. Regarding adiposity and colorectal cancer, higher levels of SAT at baseline have been linked to a better disease-free survival [30]. Chemotherapy responders have also shown an increase in their levels of adiponectin after treatment in comparison with non-responders [31]. The role of VAT may be more complex and time dependent: some studies have linked a VAT excess at baseline to lower survival [32] or more surgical complications [33], while others have found no impact [30], and some evidence points to a positive effect of VAT increase after surgery [34].

Despite its prognostic value, body composition image analysis in abdominal CT scans is not routinely conducted in many centers. To facilitate the implementation of a semi-automatic segmentation analysis in L3 axial images of CT scans in resource-limited settings, this study has compared the performances of two open-source and user-friendly software programs for body composition analysis: a picture archiving and communication system (PACS) viewer with semi-automatic HU-based threshold segmentation (Horos) and a web browser-based image analyzer with automatic threshold segmentation (CoreSlicer). Its target population is colorectal cancer due to the high prevalence of the disease and the previously described importance of body composition analysis in this group.

## 2. Materials and Methods

### 2.1. Study Design

The study design comprised an analytical observational study that was carried out in a single center (Hospital Universitario Virgen del Rocío, Seville, Spain). The measurement period was from July 2022 to June 2024.

Regarding the study sample, inclusion, exclusion, and withdrawal criteria were applied in colorectal cancer outpatients as previously published [35]. Consecutive sampling was used. Regarding the sample size, Lu et al.’s [36] methodology was applied using the statistical package blandPower (https://rdrr.io/github/nwisn/blandPower/f/README.md, accessed on 22 June 2024) [37] in Rstudio software (version 2023.06.1+524) [38]. An a priori risk of type I error = 0.05 and a risk of type II error = 0.20 were set to compare the muscle mass measurements provided by the two software programs of interest (Horos and CoreSlicer) in a Bland–Altman analysis. We predetermined a maximal clinically acceptable difference between the software programs: (δ) = 5%. A preliminary study with a subset of the study sample produced the following results: the mean of differences between Horos and CoreSlicer (μ) = 0.4 and the standard deviation of differences between Horos and CoreSlicer (σ) = 1.8. These parameters provided an estimated sample size of *n* = 68 pairs of measurements.

This study was conducted in accordance with the Declaration of Helsinki and approved by the Ethics Committee “CEI de los Hospitales Universitarios Virgen Macarena y Virgen del Rocío” (protocol code: 1006-N-22; date of approval: 23 May 2022).

### 2.2. Data Collection

#### 2.2.1. Image Analysis

Abdominal CT scans were requested by the Oncology Department of our center due to diagnostic–therapeutic reasons. Both the General Electric Revolution EVO (GE HealthCare Technologies Inc., Chicago, IL, USA) and Toshiba Aquilion (Toshiba, Minato, Japan) scanners were used. Portovenous phase scans with a slice thickness of either 1.00 or 1.25 mm were obtained after intravenous administration of a contrast medium following a standardized acquisition protocol. The images were retrospectively downloaded in Digital Imaging and Communication in Medicine (DICOM) format using our local PACS server with Philips Vue PACS (Philips, Amsterdam, The Netherlands). DICOM files were then anonymized using DICOM Anonymizer v2.4.2 (https://www.dicomanonymizer.com/index.html, accessed on 22 June 2024).

Two software programs were used in this study, both capable of tissue segmentation based on intensity thresholds. Horos is an open-source code software (FOSS) program that is distributed free of charge under the LGPL license at Horosproject.org and sponsored by Nimble Co LLC d/b/a Purview (Annapolis, MD, USA). Horos is a 64-bit medical image viewer for Mac OS X based upon OsiriX^TM^ and other open-source medical imaging libraries. Its 4.0.0RC4 version was used for this study (https://github.com/horosproject/horos/releases, accessed on 22 June 2024). CoreSlicer [39] version 1.0 is a free-of-charge, web-based, CT scan segmentator and is distributed under an MIT license. Its 1.0 version was accessed for this study (https://old.coreslicer.com, accessed on 22 June 2024) with the available source code (https://github.com/louismullie/web-ct-segmentation, accessed on 22 June 2024).

CoreSlicer has demonstrated prognostic value in rectal cancer [33] and has undergone a thorough validation process [39]. Nevertheless, Horos was selected as the gold-standard software in this study, as it has demonstrated the following requirements: prognostic capacity in the target population [40], reliability and accuracy in the measurement of myosteatosis [41], reliability and accuracy in the measurements of body composition volume [42], and excellent intra- and inter-observer agreement with itself [43] and with respect to reference software [42]. Additionally, Horos has a graphical user interface (GUI) similar to PACS viewers commonly used in Radiology as well as a 3D volumetric rendering feature that facilitates the identification of structures in patients with anatomical alterations [44].

All measurements were simultaneously performed by a single operator on a Mac Mini M1 with 16 GB of RAM (Apple Inc., Cupertino, CA, USA) and an LG 32UN500P-W 31.5-inch screen with 4K resolution (LG Electronics, Seoul, Republic of Korea). The identification of the L3 vertebra and tissue segmentation in a selected axial slice were performed in all cases following the Alberta protocol (TomoVision, Magog, QC, Canada, https://tomovision.com/Sarcopenia_Help/index.htm, accessed on 22 June 2024). The following HU thresholds were used to segment VAT (−150 to −50 HU), SAT and IMAT (−190 to −30 HU), and MT (−29 to +150 HU). We provide an example of tissue segmentation using both software programs in Figure 1. The Horos “3D Volume Rendering” function was used to three-dimensionally visualize the axial skeleton if needed.

All images were visualized with an “Abdominal CT Scan” window in both software programs. In the case of CoreSlicer, Chrome version 126.0.6478.114 (Google LLC, Menlo Park, CA, USA) was used. To increase the image size, Chrome zoom was set at 125%, and CoreSlicer zoom was set at the maximum allowable value. Analogous to van Vugt et al. [42], segmentation in Horos was initially carried out using the “Grow Region (2D/3D Segmentation)” function to select pixels according to the intensity thresholds of the Alberta protocol. In the case of CoreSlicer, the initial segmentation was carried out with the software’s “Analyze Slice” function, which includes the following intensity thresholds: −190 to −30 HU for adipose tissues (VAT and SAT) and −29 to 150 HU for MT. CoreSlicer version 1.0’s built-in algorithm includes a median filter for denoising, a threshold filter for edge detection, and a percentile filter for edge smoothing. We refer to Additional file 1 of Mullie et al. for further information [39].

The initial semi-automatic segmentation results in both programs were later manually edited by A.J.S. to ensure the anatomical accuracy of the tissues of interest. Structures erroneously included within a tissue of interest (VAT, SAT, IMAT, or MT) due to HU similarity were deselected, and pixels of the tissue of interest that would not have been included in the initial analysis were included. For this purpose, the “Brush” tool was used in Horos, while the zoom was modified as needed with the “Magnify” tool. The “Brush” tool in Horos drew or erased pixels in the ROI independently of intensity. The “UCLA” palette within the “Color Look Up Table” in Horos and multiplanar reconstruction (MPR) were used on an ad hoc basis at the discretion of the researcher (A.J.S.) to improve the identification of structures. In CoreSlicer, the brush tool (with built-in intensity thresholds depending on the tissue of interest) was used for this task. In this program, a new region called “IMAT” was created in the built-in toolbox in the right side of the screen by selecting “THRESHOLD TYPE: Fat” and then clicking on the plus icon. Segmentations were carried out simultaneously in both software programs to maximize intra-operator repeatability so that measured differences could be mainly attributable to inter-software differences.

Segmentation colors were kept the same in both software programs to facilitate a later review of high-resolution screenshots by a certified radiologist (E.S.R.). If human-made errors or inconsistencies were detected, the segmentation process was repeated, applying the necessary corrections. After image analysis was completed, the numerical values of the CSAs (cm^2^) and intensities (HU) of MT, SAT, VAT, and IMAT were later registered for both programs on an Excel spreadsheet (Microsoft Corporation, Redmond, WA, USA). Therefore, all tissue segmentation procedures were blinded to these numerical values. This information was displayed using “ROI Info” of the .roi file of each segmented tissue in Horos and the “measurements.csv” file resulting from each CoreSlicer segmentation.

#### 2.2.2. Operative Definitions of Dynapenia, Muscle Atrophy, Sarcopenia, and Visceral Obesity

The diagnosis of muscle atrophy was based on the Skeletal Muscle Index (SMI; cm^2^/m^2^), which was obtained from the measured MT CSA (cm^2^) using the formula
$$\mathrm{SMI}=(\mathrm{MT}-\mathrm{CSA})/{\mathrm{height}}^{2}$$

The diagnosis of myosteatosis was based on the measured MT intensity. The selected cutoff points for the diagnosis of both conditions were the p5 reference values for a healthy population published by Van Vugt et al. [45], as well as the prognostic thresholds provided by Dolan et al. in colorectal cancer [46]. To diagnose dynapenia, maximal handgrip strength was determined as previously described [35] on the same day of the CT scan. In this study, we used the maximal handgrip strength normative values developed by Dodds et al. [47], and dynapenia was defined with a percentile-based approach (maximal strength below the corresponding 10th percentile based on age and sex) [48]. Sarcopenia was defined as the conjunction of muscle atrophy and dynapenia using the EWGSOP-II criteria [48]. The diagnosis of visceral obesity was based on the VAT CSA (cm^2^) using the cutoff points published by Doyle et al. [49].

#### 2.2.3. Basic Anthropometry Protocol

The height and weight were measured on the same day of the CT scan following ESPEN guidance [50]. A description of the instruments used for this task is available elsewhere [35].

#### 2.2.4. Clinical Variables and Cancer Staging

Clinical variables were defined and obtained from digitized health records (“DIRAYA Clinical Station”) as previously published [35], including information about surgical treatment.

#### 2.2.5. Data Quality

All measurements were carried out by a single researcher with experience in body composition analysis (A.J.S.). Image analysis was supervised by a certified radiologist (E.S.R.) with extensive experience in abdomen imaging. Cancer stagings, treatments, and performance scores were registered in the database as recorded by oncologists (J.R.R.-M.) in health records. The type of surgery was registered in the database as recorded by surgeons (I.R.-S.) in health records.

### 2.3. Data Analysis

For the statistical analysis, the packages tidyverse [51], cowplot [52], DescTools [53], ggpubr [54], and Rcmdr [55] were used in RStudio software (version 2023.06.1+524) [38]. Normality was analyzed with the Shapiro–Wilk test. Normally distributed variables were depicted as the mean and standard deviation (SD), and non-normally distributed variables were described as the median and interquartile range (IQR). Central tendency measurements of the CSA and intensity were compared in all tissues of interest (MT, SAT, VAT, and IMAT) depending on the software program (Horos vs. CoreSlicer), both as raw measurements and relative differences (Δ) of CoreSlicer to Horos, with the latter computed as follows:Δ=(Parameter(CoreSlicer)−Parameter(Horos))/Parameter(Horos)

A *t*-test (in the presence of normality and homoscedasticity) or a Wilcoxon signed-rank test were used otherwise. Simple correlation was calculated with the Pearson correlation coefficient (*r*). Accuracy and precision regarding the CSA and intensity for each tissue of interest (MT, SAT, VAT, and IMAT) in Horos and CoreSlicer were determined using the Bland–Altman analysis [56] and Lin’s Concordance Correlation Coefficient (*ρ*), considering values > 0.99 as “near perfect”, 0.95 to 0.99 as “substantial”, 0.90 to 0.98 as “moderate”, and < 0.90 as “poor” [57]. Differences in the prevalence of sarcopenia, myosteatosis, and excess VAT were compared using an *X*^2^ test. Categorical agreement in this regard was studied using Cohen’s kappa [58]. Outliers were not censored, and all measurements were included for statistical analysis. Statistical significance was determined in all two-tailed tests as a *p*-value < 0.05.

## 3. Results

### 3.1. Clinical and Demographical Descriptions of the Study Sample

A total of *n* = 68 participants were measured and included for the analysis. Their clinical data are registered in Table 1. The study participants were mostly affected by right colon (*n* = 13) and sigmoid (*n* = 13) neoplasms. The modal TNM stage at diagnosis was IIIB (*n* = 22). A vast majority of participants had undergone surgery (*n* = 58). The most frequent type of surgery was low anterior resection (*n* = 16), followed by right hemi-colectomy (*n* = 15). Only *n* = 17 participants were under active chemotherapy at the time of measurement. The sample modal ECOG score was 1. The modal BMI was normal weight, followed by overweight. No participant displayed clinically evident signs of volume overload.

### 3.2. Image Analysis Characteristics

Regarding imaging, *n* = 63 studies were undertaken using a GE Revolution EVO scanner and *n* = 5 using a Toshiba Aquilion. All studies used intravenous contrast and had either a 1.25 mm slice thickness (*n* = 59) or 1.00 mm slice thickness (*n* = 9). The image acquisition parameters were as following: voltage = 120 kV in all cases, and amperage = 5(3) mAs. The first study was acquired on 11 July 2022 and the last on 25 April 2023.

### 3.3. Comparisons of Tissue CSAs and Tissue Intensities between Software Programs

When comparing CSAs in the tissues of interest (MT, SAT, VAT, and IMAT) using both software programs (CoreSlicer and Horos), no significant differences were found in either MT (130.682 vs. 130.852 cm^2^) or IMAT (8.700 vs. 8.317 cm^2^). Although the magnitude of the difference in IMAT was small in absolute terms (0.187 cm^2^), it proved large in relative terms (+18.045%) due to its small CSA. Regarding the other adipose tissue compartments, statistically significant differences were found in both SAT (188.911 vs. 187.352 cm^2^) and VAT (182.990 vs. 166.092 cm^2^). A detailed description of these data is in Table 2, with their graphical description in Figure 2.

When comparing the intensity in the tissues of interest (MT, SAT, VAT, and IMAT) using both software programs (CoreSlicer and Horos), all tissues presented significant differences: MT (33.237 vs. 35.398 cm^2^), SAT (−103.945 vs. −106.530 cm^2^), VAT (−87.868 vs. −92.723 cm^2^), and IMAT (−64.353 vs. −65.272 cm^2^). A detailed description of these data is in Table 3, with their graphical description in Figure 3.

### 3.4. Correlations of Tissue CSAs and Tissue Intensities between Software Programs

Pearson correlation coefficients for both the CSA and intensity in the different tissues of interest between Horos and CoreSlicer are described in Table 4. The correlation was very strong for all parameters, the strongest being the MT CSA (*r* = 0.998; IC95%: 0.997 to 0.998), SAT CSA (*r* = 0.998; IC95%: 0.997 to 0.999), and VAT CSA (*r* = 0.998; IC95%: 0.996 to 0.998). The IMAT intensity had the weakest correlation (*r* = 0.843; IC95%: 0.756 to 0.900). The graphical representation of these data is depicted in Figure 4.

### 3.5. Agreement of Tissue CSA and Tissue Intensity between Software Programs

The complete agreement analysis for the CSA using both software programs (Horos and CoreSlicer) is available in Table 5. *ρ* was near perfect in the MT and SAT and substantial in the VAT and IMAT. Regarding the Bland–Altman analysis—and both in absolute and relative terms—the MT had the smallest bias (+0.225 cm^2^; Δ = 0.188%), while VAT (+13.052 cm^2^; Δ = 11.881%) and SAT (+4.456 cm^2^; Δ = 3.479%) displayed the greatest biases. All LoAs were moderately wide, except for those of MT (−3.354 to 3.804 cm^2^, range = 7.159 cm^2^) and IMAT (−2.432 to 3.124 cm^2^, range = 5.557 cm^2^). The Bland–Altman analyses for the different tissues of interest are graphically represented in Figure 5.

The complete agreement analysis for the intensity using both software programs (Horos and CoreSlicer) is available in Table 6. *ρ* was substantial for MT, moderate for SAT, and poor for VAT and IMAT. In the Bland–Altman analysis, all analyzed tissues presented bias. Both in absolute and relative terms, VAT had the greatest bias (+4.6 HU; Δ = −5.582%), and IMAT the smallest bias (+1.1 HU, Δ = −1.522%). All LoAs were moderately wide except for MT (−4.526 to 1.635 HU, range = 6.162 HU). The Bland–Altman analyses for the different tissues of interest are graphically represented in Figure 6.

### 3.6. Handgrip Strength and Dynapenia

Handgrip strength followed a normal distribution at 34.7 (10.6) kg. Based on their handgrip strength, *n* = 3 participants had dynapenia. There was no colocalization for dynapenia and muscle atrophy, and therefore, no diagnosis of sarcopenia was made in the study sample.

### 3.7. Misclassification Error and Its Clinical Impact

SMI was non-normally distributed (*p*-value = 3.812 × 10^−3^) in the study sample. SMI measurements using CoreSlicer and Horos were as follows: 47.484 (13.838) vs. 46.999 (13.931) cm^2^/m^2^, with no statistically significant differences. These data are represented in Figure 7.

Using Van Vugt et al.’s cutoff points [45], both CoreSlicer and Horos diagnosed *n* = 1 cases of muscle atrophy, displaying the exact same prevalence (1.47%) of this condition. The participant was a 69-year-old man, his SMI being 39.425 cm^2^/m^2^ in CoreSlicer and 39.728 cm^2^/m^2^ in Horos. Using Dolan et al.’s cutoff points [46], both CoreSlicer and Horos diagnosed *n* = 34 cases of muscle atrophy, displaying the exact same prevalence (50.00%) of this condition. Agreement was almost perfect in both cases, with *κ* = 1.000.

Using Van Vugt et al.’s cutoff points [45], there were *n* = 8 cases (11.76% prevalence) of myosteatosis using CoreSlicer, while Horos diagnosed *n* = 4 cases of myosteatosis (5.88% prevalence) in the study sample. These differences (+5.88%) were not statistically significant. Agreement was substantial in this case, with *κ* = 0.638 (*p*-value = 1.64 × 10^−8^). Using Dolan et al.’s cutoff points [46], there were *n* = 32 cases (47.05% prevalence) of myosteatosis using CoreSlicer, while Horos diagnosed *n* = 26 cases of myosteatosis (38.23% prevalence) in the study sample. These differences (+8.82%) were not statistically significant. Agreement was almost perfect in this case, with *κ* = 0.821 (*p*-value = 5.92 × 10^−12^).

Using Doyle et al.’s cutoff points [49], there were *n* = 47 cases (69.11% prevalence) of visceral obesity using CoreSlicer, while Horos diagnosed *n* = 45 cases of visceral obesity (66.17% prevalence) in the study sample. These differences (+2.94% prevalence) were not statistically significant. Agreement was near perfect in this case, with *κ* = 0.821 (*p*-value = 5.92 × 10^−12^).

## 4. Discussion

Both software programs (Horos and CoreSlicer) proved to be feasible tools for body composition analysis by a non-radiologist M.D. using abdominal CT scans in the study sample. In general, the LoAs in this study are in line with previous studies [41,59,60] that compared either Horos or OsiriX to other human-supervised tissue segmentation software programs. The creation of a new IMAT region in CoreSlicer produced similar and accurate results regarding the IMAT CSA, albeit the IMAT intensity displayed significant differences and poor agreement between software programs.

The compared MT CSA measurements had good precision and accuracy, so both CoreSlicer and Horos displayed the exact same cases of muscle atrophy under the operative definitions of Van Vugt et al. and Dolan et al. Interestingly, the prevalence of muscle atrophy varied greatly depending on the definition—due to the intrinsic characteristics of the cutoff points—yet inter-software agreement was perfect in both cases. Barbalho et al. also found no muscle atrophy misclassification using Slice-O-Matic and OsiriX [59].

Probably due to its significantly lower MT intensity, there was an increase in the prevalence of myosteatosis using CoreSlicer, although its clinical impact is debatable. This happened under the operative definitions of both Van Vugt et al. [45], and Dolan et al. [46]. Nevertheless, it did not reach statistical significance, and the agreement between the software programs for the diagnosis of myosteatosis was substantial or near perfect. Regarding adipose tissue, we found an inaccuracy and overestimation of both the CSA and intensity in adipose tissues (primarily VAT, followed by SAT) with CoreSlicer. Although concordance was poor for the VAT and SAT intensities, concordance was near perfect for the SAT CSA and substantial for the VAT CSA. Despite these differences, the prevalence of abdominal obesity was very similar using Doyle et al.’s cutoff points [49]. We think that caution should be applied when trying to extrapolate the MT intensity and SAT or VAT results between programs. Therefore, adhering to a single program (either CoreSlicer or Horos) to measure these parameters could be an adequate measure to ensure internal consistency in future study protocols.

A recent study by Viddeleer et al. [61] demonstrated that CoreSlicer and OsiriX (the non-freeware forefather of Horos) could produce the exact same measurements for both the CSAs and intensities of MT, VAT, and SAT in a phantom DICOM that schematically represented these tissues in an axial slice of an abdominal CT scan. These measurements were also identical to those generated using gold-standard software (Slice-o-Matic). Our results slightly differ from these findings, but we think this could be attributable to differences in the study design. Firstly, highly trained radiologists participated in the previous study, and a single image was analyzed. In our study, a non-radiologist scouted 68 abdominal CT scans to find the adequate location and then separately segmented the selected slices in both software programs. A total of 136 independent processes were performed, and all images were analyzed independently of the presence of complex anatomical structures or post-surgical modifications. These real-life conditions may have enhanced software-dependent differences in image visualization (due to differences in the graphical user interface, spatial resolution, and image positioning) and segmentation (characteristics of the built-in algorithms, brush sizes, and the presence of an HU-based brush threshold), further modifying the human-supervised segmentation process.

To our knowledge, this is the first study that cross-sectionally compares Horos and CoreSlicer in a real-life setting, exploring their measuring bias and the diagnostic impact of each tool in a colorectal cancer sample. All measured individuals were selected for statistical analysis independently of extreme values or the presence of anatomical alterations such as ostomy or other abdominal wall surgical sequelae, increasing the validity of the study. Although the statistical analysis was not blinded, numerical data were only analyzed after the completion of the image analysis. The radiologist reviewer was blinded to the type of software that produced the analysis, and the research team had no conflicts of interest. Finally, handgrip strength measurements and basic anthropometry were performed on the same day of the abdominal CT scan, increasing the data quality and providing a solid diagnostic base for dynapenia and sarcopenia.

Our study also has several limitations. Its main observer in our study is a certified endocrinologist who has been supervised by a certified radiologist, although previous studies have demonstrated that trained non-radiologist healthcare professionals can produce precise and consistent results [62]. When making comparisons with other studies, it should be considered that the CSAs of the tissues of interest have been measured in portovenous phase. Our previous experience (not published) with CoreSlicer leads us to emphasize that a 31.5-inch 4K resolution screen was used: due to a more limited visualization than a typical PACS viewer, the use of smaller screens may reduce the quality of the analysis. Although slight differences in the segmentation time or ease of use of each software were found, they have not been included in the Results section, as they have not been objectively quantified. It is also important to note that a 2.0 version of CoreSlicer is in progress, with a presumably upgraded AI segmentation tool in comparison with the non-AI-based 1.0 version used in this study. Finally, a Jaccard or Sorensen–Dice index has not been performed to enhance the comparison of the segmentation results between the software programs.

## 5. Conclusions

Both Horos and CoreSlicer were feasible segmentation software programs in the present study. The MT CSA showed great inter-software agreement and no muscle atrophy misdiagnosis. Segmentation differences in the MT intensity and VAT CSA caused limited diagnostic misclassification in the study sample.

## Figures and Tables

**Figure 1 diagnostics-14-01696-f001:**
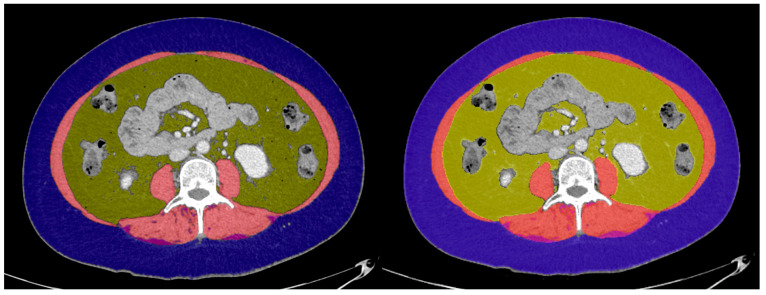
Examples of tissue segmentation using Horos (**left**) and CoreSlicer (**right**) in the same slice at the third lumbar vertebra in an abdominal CT scan. The following tissues were segmented: muscle tissue (MT, represented in red), subcutaneous adipose tissue (SAT, represented in blue), visceral adipose tissue (VAT, represented in yellow), and intramuscular adipose tissue (IMAT, represented in purple).

**Figure 2 diagnostics-14-01696-f002:**
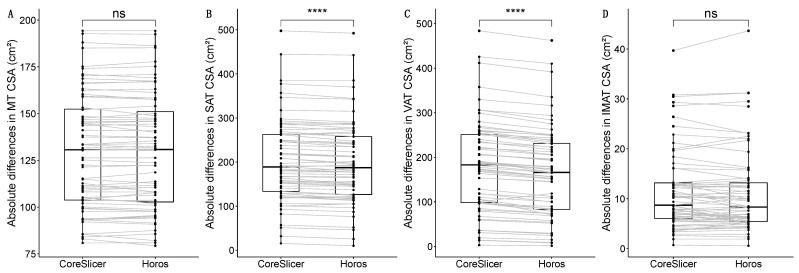
Comparisons of the CSA (Cross-Sectional Area) in cm^2^ for the different tissues of interest between software programs (Horos and CoreSlicer): muscle tissue (MT) (**A**); subcutaneous adipose tissue (SAT) (**B**); visceral adipose tissue (VAT) (**C**); intramuscular adipose tissue (IMAT) (**D**). The significance of the performed Wilcoxon signed-rank test appears as either “ns” (not significant) or with the following symbols representing *p*-values: **** < 0.0001.

**Figure 3 diagnostics-14-01696-f003:**
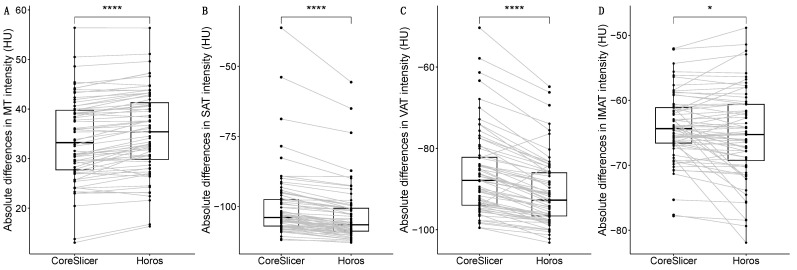
Comparisons of intensity in HU (Hounsfield Units) for the different tissues of interest between software programs (Horos and CoreSlicer): muscle tissue (MT) (**A**); subcutaneous adipose tissue (SAT) (**B**); visceral adipose tissue (VAT) (**C**); intramuscular adipose tissue (IMAT) (**D**). The significance of the performed Wilcoxon signed-rank test appears as either “ns” (not significant) or with the following symbols representing *p*-values: * < 0.05; **** < 0.0001.

**Figure 4 diagnostics-14-01696-f004:**
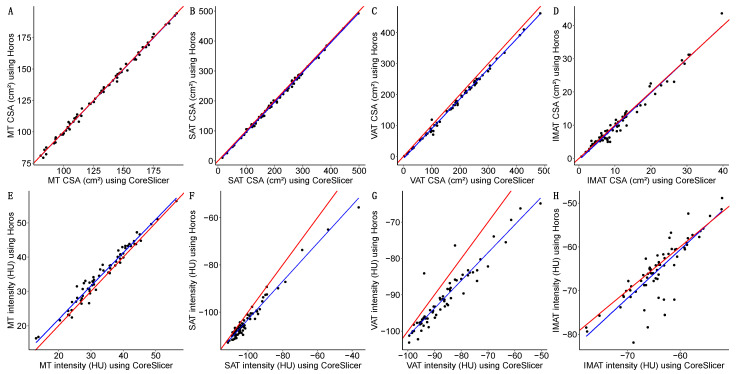
Simple linear regression of the CSA (Cross-Sectional Area) in cm^2^ for the different tissues of interest between software programs (Horos and CoreSlicer): muscle tissue (MT) (**A**); subcutaneous adipose tissue (SAT) (**B**); visceral adipose tissue (VAT) (**C**); intramuscular adipose tissue (IMAT) (**D**). Simple linear regression of the intensity in HU (Hounsfield Units) for the different tissues of interest between software programs (Horos and CoreSlicer): muscle tissue (MT) (**E**); subcutaneous adipose tissue (SAT) (**F**); visceral adipose tissue (VAT) (**G**); intramuscular adipose tissue (IMAT) (**H**). In all cases, the perfect bisector of inter-software regression is shown as a solid red line, and the linear regression model between the software programs is shown as a solid blue line. Please note the different scales between graphics.

**Figure 5 diagnostics-14-01696-f005:**
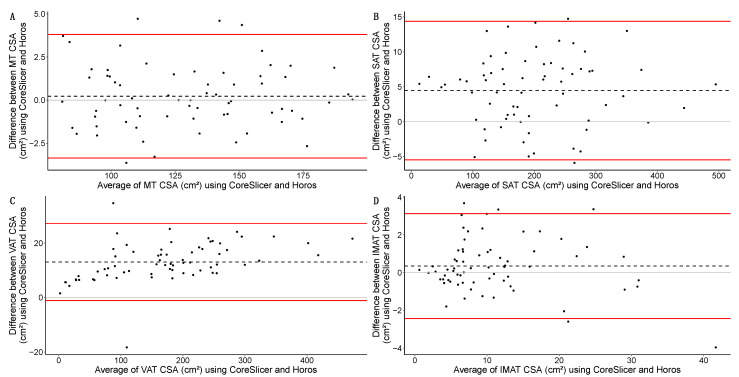
Bland–Altman analyses of the CSA (Cross-Sectional Area) in cm^2^ for the different tissues of interest between software programs (Horos and CoreSlicer): muscle tissue (MT) (**A**); subcutaneous adipose tissue (SAT) (**B**); visceral adipose tissue (VAT) (**C**); intramuscular adipose tissue (IMAT) (**D**) in the whole study sample. Averages of the CSA for both software programs are presented on the *X*-axis; differences in the CSA for both software programs for both software programs are presented on the *Y*-axis. Individual measurements are shown as circles; biases are represented as black dashed lines; upper and lower limits of agreement are shown as red solid lines. The absence of differences (*Y* intercept = 0) between the software programs (Horos and CoreSlicer) is represented as a grey line. Please note the different scales across graphics.

**Figure 6 diagnostics-14-01696-f006:**
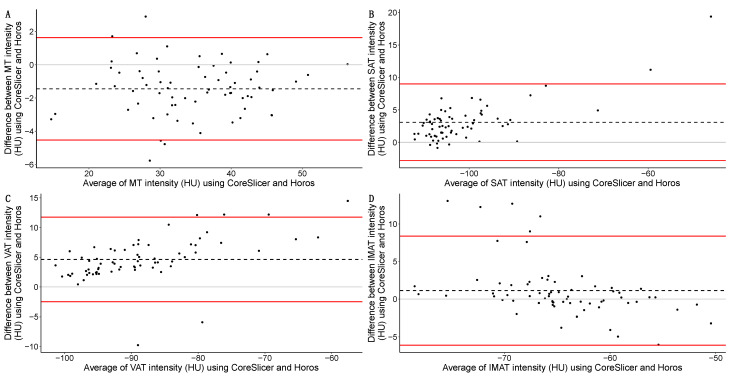
Bland–Altman analyses of the intensity in HU (Hounsfield Units) for the different tissues of interest between software programs (Horos and CoreSlicer): muscle tissue (MT) (**A**); subcutaneous adipose tissue (SAT) (**B**); visceral adipose tissue (VAT) (**C**); intramuscular adipose tissue (IMAT) (**D**) in the whole study sample. Averages of the CSA for both software programs are presented on the *X*-axis; differences in the CSA for both software programs for both software programs are presented on the *Y*-axis. Individual measurements are shown as circles; biases are represented as black dashed lines; upper and lower limits of agreement are shown as red solid lines. The absence of differences (*Y* intercept = 0) between the software programs (Horos and CoreSlicer) is represented as a grey line. Please note the different scales across graphics.

**Figure 7 diagnostics-14-01696-f007:**
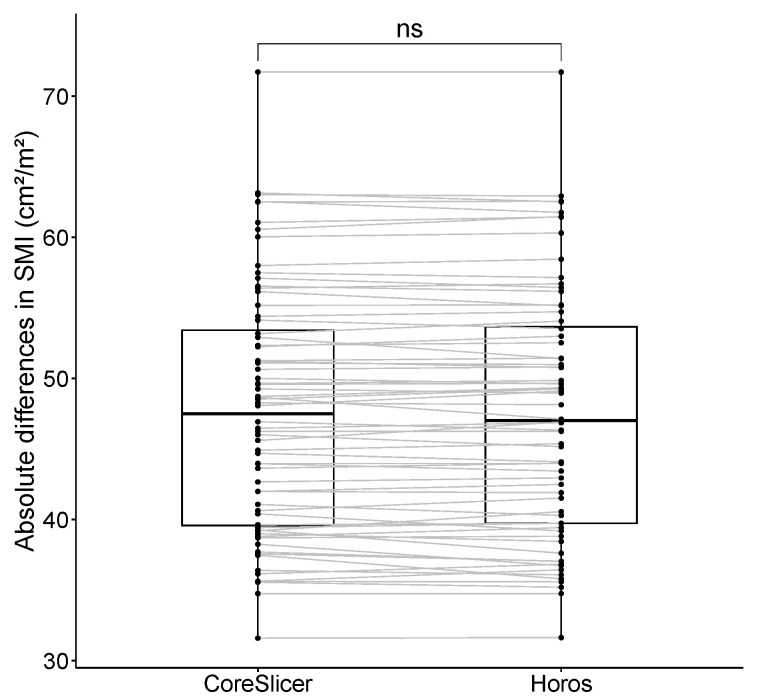
Comparisons of the SMI (Skeletal Muscle Index) in cm^2^/m^2^ between software programs (Horos and CoreSlicer). The significance of the performed Wilcoxon signed-rank test appears as “ns” (not significant).

**Table 1 diagnostics-14-01696-t001:** Clinical and demographic characteristics of the study sample.

Parameter	Results
Sample size (*n_i_*)	*n* = 68
Age (years)	Me = 64.72
	IQR = 12.67
Older than 65 (*n_i_*)	*n* = 32 (47.05%)
Female (*n_i_*)	*n* = 31 (45.58%)
Neoplasm location (*n_i_*)	Right colon, *n* = 13
	Transverse colon, *n* = 4
	Left colon, *n* = 6
	Rectosigmoid, *n* = 5
	Sigma, *n* = 13
	Rectum, *n* = 27
Stage (TNM) at diagnosis	IIA (*n* = 10); IIB (*n* = 2); IIC (*n* = 2)
	IIIA (*n* = 4); IIIB (*n* = 22); IIIC (*n* = 9)
	IVA (*n* = 9); IVB (*n* = 10); IVC (*n* = 0)
Previous surgery	Yes, *n* = 58
	No, *n* = 10
First surgery	Abdomino-perineal resection, *n* = 3
	Colostomy, *n* = 2
	Hepatectomy, *n* = 3
	Left hemi-colectomy, *n* = 8
	Low anterior resection, *n* = 16
	Right hemi-colectomy, *n* = 15
	Sigmoidectomy, *n* = 10
	Sub-total colectomy, *n* = 1
Active chemotherapy (*n_i_*)	Yes, *n* = 17
	No, *n* = 51
ECOG (*n_i_*)	0, *n* = 46
	1, *n* = 22
Weight (kg)	Mean = 74.17
	SD = 14.61
Height (m)	Mean = 1.644
	SD = 0.092
BMI (kg/m^2^)	Me = 27.0
	IQR = 4.6
BMI by group (*n_i_*)	Underweight, *n* = 4
	Normal weight, *n* = 20
	Overweight, *n* = 30
	Grade 1 obesity, *n* = 7
	Grade 2 obesity, *n* = 7

Clinical characteristics of the sample. Numerical values are expressed as absolute frequencies (*n_i_*), percentages (%), medians (Me), and interquartile ranges (IQRs).

**Table 2 diagnostics-14-01696-t002:** Comparisons in the CSA (cm^2^) for the different tissues using Horos and CoreSlicer.

	CoreSlicer	Horos	Absolute Differences (cm^2^)	Δ (%)	*p*-Value
MT	130.682 (48.583)	130.852 (48.260)	−0.008 (2.334)	−0.007 (1.882)	0.537
SAT	188.911 (128.498)	187.352 (131.454)	5.349 (6.844)	+2.576 (4.702)	2.3 × 10^−8^
VAT	182.990 (152.620)	166.092 (147.960)	12.171 (8.815)	+8.624 (5.591)	7.9 × 10^−12^
IMAT	8.700 (7.112)	8.317 (7.760)	0.187 (1.556)	+18.045 (2.572)	0.08

CSA: Cross-Sectional Area; Δ = (CoreSlicer CSA − Horos CSA) ÷ Horos CSA; MT: muscle tissue; SAT: subcutaneous adipose tissue; VAT: visceral adipose tissue; IMAT: intramuscular adipose tissue.

**Table 3 diagnostics-14-01696-t003:** Comparisons of the intensity (HU) for the different tissues using Horos and CoreSlicer.

	CoreSlicer	Horos	Absolute Differences (HU)	Δ (%)	*p*-Value
MT	33.237 (12.038)	35.398 (11.421)	−1.388 (2.016)	−4.163 (5.226)	4.4 × 10^−9^
SAT	−103.945 (9.417)	−106.530 (8.177)	2.538 (2.695)	−2.459 (2.688)	2.8 × 10^−12^
VAT	−87.868 (11.788)	−92.723 (10.624)	4.141 (3.824)	−4.542 (4.255)	7.8 × 10^−11^
IMAT	−64.353 (5.453)	−65.272 (8.659)	0.368 (2.201)	−0.537 (3.362)	0.026

HU: Hounsfield Units; Δ = (CoreSlicer HU − Horos HU) ÷ Horos HU; MT: muscle tissue; SAT: subcutaneous adipose tissue; VAT: visceral adipose tissue; IMAT: intramuscular adipose tissue.

**Table 4 diagnostics-14-01696-t004:** Pearson correlation coefficients (*r*) for the CSA and intensity in the different tissues using Horos and CoreSlicer.

Measured Tissue	CSA (cm^2^, 95% CI)	Intensity (HU, 95% CI)
MT	0.998 (0.997 to 0.998)	0.982 (0.971 to 0.989)
SAT	0.998 (0.997 to 0.999)	0.984 (0.975 to 0.990)
VAT	0.998 (0.996 to 0.998)	0.946 (0.914 to 0.966)
IMAT	0.985 (0.976 to 0.990)	0.843 (0.756 to 0.900)

95% CI: 95% confidence interval; CSA: Cross-Sectional Area; MT: muscle tissue; SAT: subcutaneous adipose tissue; VAT: visceral adipose tissue; IMAT: intramuscular adipose tissue. The 95% confidence intervals are represented in parenthesis.

**Table 5 diagnostics-14-01696-t005:** Agreement analysis for the CSA in the different tissues using Horos and CoreSlicer.

	MT	SAT	VAT	IMAT
*ρ* (95% CI)	0.998 (0.997; 0.998)	0.997 (0.995; 0.998)	0.989 (0.984; 0.992)	0.984 (0.974; 0.990)
Bias (cm^2^)	0.225	4.456	13.052	0.346
Δ (%)	0.188	3.479	11.881	4.259
Lower LoA (cm^2^)	−3.354	−5.460	−1.151	−2.432
Upper LoA (cm^2^)	3.804	14.374	27.256	3.124
LoA range (cm^2^)	7.159	19.835	28.407	5.557

*ρ*: Concordance Correlation Coefficient; CI: confidence interval; CSA: Cross-Sectional Area; MT: muscle tissue; SAT: subcutaneous adipose tissue; VAT: visceral adipose tissue; IMAT: intramuscular adipose tissue, LoA: Limit of Agreement.

**Table 6 diagnostics-14-01696-t006:** Agreement analysis for the intensity in the different tissues using Horos and CoreSlicer.

	MT	SAT	VAT	IMAT
*ρ* (95% CI)	0.967 (0.949; 0.979)	0.930 (0.902; 0.951)	0.824 (0.755; 0.875)	0.797 (0.706; 0.862)
Bias (HU)	−1.445	3.078	4.628	1.130
Δ (%)	−4.560	−3.537	−5.582	−1.522
Lower LoA (HU)	−4.526	−2.836	−2.493	8.377
Upper LoA (HU)	1.635	8.993	11.750	−6.115
LoA range (HU)	6.162	11.829	14.244	14.493

*ρ*: Concordance Correlation Coefficient; CI: confidence interval; HU: Hounsfield Units; MT: muscle tissue; SAT: subcutaneous adipose tissue; VAT: visceral adipose tissue; IMAT: intramuscular adipose tissue; LoA: Limit of Agreement.

## Data Availability

The data presented in this study are available on request from the corresponding author as per European legislation on data protection.

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
