# Peer review of "A Cross-Sectional Validation of Horos and CoreSlicer Software Programs for Body Composition Analysis in Abdominal Computed Tomography Scans in Colorectal Cancer Patients"

_diagnostics, 2024, doi:10.3390/diagnostics14151696_

Round 1

Reviewer 1 Report

Comments and Suggestions for Authors

The manuscript is well written and organized but it has some queries to be answer: 

1. The study was conducted at a single center with a relatively small sample size of 68 participants. This limits the generalizability of the findings to broader populations and other clinical settings.

2. The segmentation process relied on a single observer with experience in body composition analysis. Despite supervision by a certified radiologist, the potential for observer bias and variability exists.

3. The study used specific versions of the Horos and CoreSlicer software. As software evolves, newer versions may offer improved functionality and accuracy, making the findings potentially outdated quickly.

4. Differences in graphical user interfaces, spatial resolution, and segmentation tools between the two software programs may influence the segmentation outcomes. CoreSlicer’s tendency to overestimate VAT and SAT could lead to diagnostic inconsistencies.

5. While the radiologist reviewing the high-resolution screenshots was blinded to the software type, the initial image analysis was not blinded, which could introduce bias in the results.

6. The study’s setting, including the use of a high-resolution 31.5-inch 4K screen, might not reflect the conditions in many clinical environments, potentially limiting the practical application of the findings.

7. Despite overall feasibility, the study noted limited diagnostic misclassification, particularly in muscle tissue intensity and VAT CSA. This could impact clinical decisions if not addressed in practical applications.

Author Response

Dear colleague, please see the attachment. Thank you.

Reviewer 2 Report

Comments and Suggestions for Authors

Such comparisons are meaningful in diagnostics, offering colleagues who cannot afford expensive commercial software an alternative solution.

However, there are some issues in the article that need to be clarified. Firstly, CoreSlicer has been divided into versions 1.0 and 2.0. From version 2.0 onwards, it seems that it is no longer distributed as open-source software; instead, it operates on AI servers using artificial intelligence for organizational segmentation. According to the author, the use of CoreSlicer's AI segmentation feature likely pertains to version 2.0. At this point, it loses a benchmark compared with Heros (Heros analyzes through traditional image processing and threshold methods and is under the MIT license).

If the text serves as a summary of "Comparison of Two Open Source Software" as described, the author must clearly identify the differences between them and specify how to access the software (GitHub URL); otherwise, it merely becomes an advertisement for CoreSlicer.

Author Response

(The authors gave the same response as above.)

Round 2

Reviewer 2 Report

Comments and Suggestions for Authors

OK